# Inappropriate antibiotic use in the COVID-19 era: Factors associated with inappropriate prescribing and secondary complications. Analysis of the registry SEMI-COVID

Jorge Calderón-Parra[1]*, Antonio Muiño-Miguez[2], Alejandro D. Bendala-Estrada[2], Antonio Ramos-Martínez[1], Elena Muñez-Rubio[1], Eduardo Fernández Carracedo[2], Javier Tejada Montes[3], Manuel Rubio-Rivas[4], Francisco Arnalich-Fernandez[5], Jose Luis Beato Pérez[6], Jose Miguel García Bruñén[7], Esther del Corral Beamonte[8], Paula Maria Pesqueira Fontan[9], Maria del Mar Carmona[10], Rosa Fernández-Madera Martínez[11], Andrés González García[12], Cristina Salazar Mosteiro[13], Carlota Tuñón de Almeida[14], Julio González Moraleja[15], Francesco Deodati[16], María Dolores Martín Escalante[17], María Luisa Asensio Tomás[18], Ricardo Gómez Huelgas[19], José Manuel Casas Rojo[16], Jesús Millán Núñez-Cortés[2], for the SEMI-COVID-19 Network

1 Infectious Diseases Unit, Internal Medicine Department, Puerta de Hierro University Hospital, Majadahonda, Madrid, Spain, 2 Internal Medicine Department, Gregorio Marañón University Hospital, Madrid, Spain, 3 Internal Medicine Department, 12 de Octubre University Hospital, Madrid, Spain, 4 Internal Medicine Department, Bellvitge University Hospital-IDIBELL, L'Hospitalet de Llobregat, Barcelona, Spain, 5 Internal Medicine Department, La Paz University Hospital, Madrid, Spain, 6 Internal Medicine Department, Albacete University Hospital Complex, Albacete, Spain, 7 Internal Medicine Department, Miguel Servet Hospital, Zaragoza, Spain, 8 Internal Medicine Department, Royo Villanova Hospital, Zaragoza, Spain, 9 Internal Medicine Department, Santiago Clinical Hospital, Santiago de Compostela, A Coruña, Spain, 10 Internal Medicine Department, Dr. Peset University Hospital, Valencia, Spain, 11 Internal Medicine Department, Cabueñes Hospital, Gijón, Asturias, Spain, 12 Systemic Autoimmune Diseases and Rare Diseases Unit, Internal Medicine Department, Ramón y Cajal University Hospital, IRYCIS, Madrid, Spain, 13 Internal Medicine Department, Nuestra Señora del Prado Hospital, Talavera de la Reina, Toledo, Spain, 14 Internal Medicine Department, Zamora Hospital Complex, Zamora, Spain, 15 Internal Medicine Department, Virgen de la Salud Hospital, Toledo, Spain, 16 Internal Medicine Department, Infanta Cristina University Hospital, Parla, Madrid, Spain, 17 Internal Medicine Department, Costa del Sol Hospital, Marbella, Málaga, Spain, 18 General Internal Medicine Department, San Juan de Alicante University Hospital, San Juan de Alicante, Alicante, Spain, 19 Internal Medicine Department, Regional University Hospital of Málaga, Biomedical Research Institute of Málaga (IBIMA), University of Málaga (UMA), Málaga, Spain

* jorge050390@gmail.com

## Abstract

### Background

Most patients with COVID-19 receive antibiotics despite the fact that bacterial co-infections are rare. This can lead to increased complications, including antibacterial resistance. We aim to analyze risk factors for inappropriate antibiotic prescription in these patients and describe possible complications arising from their use.

### Methods

The SEMI-COVID-19 Registry is a multicenter, retrospective patient cohort. Patients with antibiotic were divided into two groups according to appropriate or inappropriate

**Data Availability Statement:** All relevant data are within the manuscript and its Supporting Information files.

**Funding:** The authors recieved no specific funding for this work.

**Competing interests:** The authors have declared that no competing interests exist.

prescription, depending on whether the patient fulfill any criteria for its use. Comparison was made by means of multilevel logistic regression analysis. Possible complications of antibiotic use were also identified.

## Results

Out of 13,932 patients, 3047 (21.6%) were prescribed no antibiotics, 6116 (43.9%) were appropriately prescribed antibiotics, and 4769 (34.2%) were inappropriately prescribed antibiotics. The following were independent factors of inappropriate prescription: February-March 2020 admission (OR 1.54, 95%CI 1.18–2.00), age (OR 0.98, 95%CI 0.97–0.99), absence of comorbidity (OR 1.43, 95%CI 1.05–1.94), dry cough (OR 2.51, 95%CI 1.94–3.26), fever (OR 1.33, 95%CI 1.13–1.56), dyspnea (OR 1.31, 95%CI 1.04–1.69), flu-like symptoms (OR 2.70, 95%CI 1.75–4.17), and elevated C-reactive protein levels (OR 1.01 for each mg/L increase, 95% CI 1.00–1.01). Adverse drug reactions were more frequent in patients who received ANTIBIOTIC (4.9% vs 2.7%, $p < .001$).

## Conclusion

The inappropriate use of antibiotics was very frequent in COVID-19 patients and entailed an increased risk of adverse reactions. It is crucial to define criteria for their use in these patients. Knowledge of the factors associated with inappropriate prescribing can be helpful.

## Introduction

Since the beginning of 2020, the world has faced the threat posed by the coronavirus disease 2019 (COVID-19) pandemic. As of March 12th, more than 110 million people have been infected and more than 2 million people have died worldwide [1].

During the first wave, it has been observed that most patients admitted with a COVID-19 had been prescribed antibiotics, including broad-spectrum antibiotics in a percentage of cases. Antibiotic use has been described in more than 70% of cases [2, 3]. Suspicion of concomitant bacterial pneumonia and evidence of superinfection may have been a motivating factor behind this extensive use. However, some studies suggest that bacterial co-infection is rare, occurring in less than 10% of cases [4, 5]. More recent literature have confirmed that bacterial co-infection and super-infection is rare, representing 8.5–12% of cases [3, 6]. Inappropriate antibiotic prescribing in COVID-19 patients can lead to avoidable complications, including increased bacterial resistance [7], *Clostridioides difficile* (CD) infection [8] reactions, renal impairment, and more. All of the above negative repercussions are possible and yet no benefits to patients have been described [9]. Therefore, several groups have sounded the alarm and requested the intervention of antibiotic stewardship programs in these patients [10].

We aim to analyze systemic inappropriate antibiotic prescribing in patients with SARS--CoV-2 infection in order to determine the proportion of patients who were inappropriately prescribed antibiotics as well as to identify the factors associated with unjustified treatment. This work also aims to describe the possible complications arising from antibiotic prescription. The primary outcome was the proportion of inappropriate antibiotic and its risk factors comparing to appropriate antibiotic. Secondary outcomes included risk factor for inappropriate prescription vs no antibiotic use, complications from antibiotic prescription and compare inappropriate prescription during the study period.

## Patients and methods

The SEMI-COVID-19 Registry is an ongoing retrospective observational cohort study that includes consecutive patients who were discharged after hospitalization or died due to COVID-19 in 150 hospitals in Spain from March 1, 2020 on. This work analyzed data collected up to June 23, 2020.

### Study population and participants

The inclusion criteria for this study were: a) patients 18 years of age or older, b) confirmed COVID-19 diagnosis, c) first hospital admission to a Spanish hospital participating in the registry, d) discharge from the hospital or in-hospital death, and e) informationon antibiotic use during hospitalization available. COVID-19 was confirmed by a positive real-time polymerase chain reaction (PCR) test of a nasopharyngeal exudate sample, sputum, or bronchoalveolar wash or by a positive result on a serological and a compatible clinical presentation. Patients could be included in the registry with a first negative PCR if subsequent determination in the other samples was positive. The exclusion criteria were hospital readmissions of the same patient or absence of informed consent. Patients were treated at the discretion of their attending physician.

### Ethical consideration

Personal data were processed in compliance with Law 14/2007 of July 3, Biomedical Research, as well as Regulation EU 2016/679 of the European Parlament and of the Council of 27 April 2016, General Data Protection Regulation and Organic Law 3/2018 of 5 December on the Protection of Personal Data and Guarantee of Digital Rights. The registry has the approval of the Ethics and Research Committee of the Province of Malaga. The Department of Medicinal Products for Human Use of the Spanish Medicines and Healthcare Products has classified the study as "Non-Post-Authorization Observational Study". Patients were asked for a written informed consent during hospital admission. Due to biosecurity reasons, the consent had not witnessed. When it was not possible to obtain it for biosecurity reasons or because the patient was already discharged, it was collected verbally, leaving evidence in their medical history.

### Registry information and definitions

The methods of this registry have been fully described in previously published works [11]. In summary, all consecutive patients who were discharged after March 1 on hospitals participating in the register were included. Data were collected anonymously and retrospectively by local investigators in each center. The data collected include approximately 300 variables grouped under several headings. Due to the characteristics of the database, it was not possible to analyze the specific antibiotic prescribed within the different antibiotic families nor was it possible to analyze the duration of treatment or the time it was started.

In order to classify antibiotic prescribing, the following criteria of appropriate prescribing were considered: negative SARS-CoV-2 PCR (in a scenario of a patient admitted with pneumonia without confirmed COVID-19, the empirical use of antibiotics until COVID-19 confirmation could be justified), shock/sepsis, clinical symptoms, radiological findings or laboratory test suggestive of bacterial superinfection, including purulent expectoration, unilateral alveolar (with air bronchogram) infiltrate, significant pleural effusion, CT imaging that is not compatible with COVID-19, and procalcitonin (PCT) equal to or greater than 0.5 ng/mL, and confirmed bacterial complications, including respiratory bacterial coinfection (at admission time) or superinfection (later on admission) with microbial isolation, urinary tract infection, abdominal infections, skin and soft tissue infection, and other infections. PCT elevation has been

associated with bacterial superinfection in COVID-19 patients [12, 13], and some authors suggest its use to guide antibiotic initiation in these patients [14]. Thus, prescribing was considered appropriate when patients who met any of these criteria received antibiotic treatment. These criteria for the appropriate use of antibiotics are similar to those proposed in the literature by several authors [15–17]. During the first months of the pandemic, early evidence suggested that macrolides could have inhibitory action on SARS-CoV-2 and immunomodulatory effects on patients [18–20]. Accordingly, many local protocols in our country recommended the use of macrolides in COVID-19 patients due to these effects, and not as an antibacterial drug. We provide examples local protocols including this recommendation in S1 Annex. Therefore, we decided to categorize patients who were prescribed macrolides without any other antibiotic drug as patients with no antibiotic prescription, since we consider that, in these patients, macrolides were not used for their antibiotic effect, but as an immunomodulatory and antiviral drug (comparable to lopinavir-ritonavir or hydroxychloroquine use).

Among the various entities included in the "other complications" variable in the registry (a variable which was based on a free text), we manually identified the following complications which could be potentially associated with antibiotic use: pharmacological hypertransaminasemia, drug-induced diarrhea, rash/allergy caused by antibiotics, CD diarrhea, invasive and non-invasive candidiasis, QT prolongation, drug-induced neutropenia and drug-induced thrombocytopenia. We defined flu-like symptoms as the presence of odynophagia, myalgia, arthralgia, headache, or asthenia.

## Study management

The promoter of this study is the Spanish Society of Internal Medicine (SEMI). The researchers who coordinated the study at each hospital agreed to participate in the study voluntarily and without remuneration. The monitoring of the study is carried out by the SEMI scientific committee and an independent agency.

## Statistical analysis

Demographic, clinical, epidemiological, laboratory and diagnostic imaging data of the participating patients were analyzed. Quantitative variables were expressed as median (interquartile range (IQR)). Categorical variables were expressed as absolute frequencies and percentages. For univariant analysis, the chi-squared test was used for qualitative variables (or Fisher's exact test when necessary) and the Student's t-test for quantitative variables (or Wilcoxon W when necessary). Variables that achieved statistically significant and clinically relevant differences in the univariant analysis were included in a single-step multivariate logistic regression analysis model. Corrected *odds ratio* (OR) and 95% confidence intervals (CI) for inappropriate prescription were provided for all the included variables. Bilateral p-values below 0.05 were considered significant. Statistical analysis was performed using the SPSS version 25 software package (IBM SPSS Statistics for Windows, Version 25.0. Armonk, NY: IBM Corp).

## Results

A total of 14,907 patients had been included in the registry as of June 23, 2020. All data necessary for inclusion were available on 13,932 patients. Fig 1 shows the patient flowchart.

## Antibiotics prescription

Of these 13,932 patients, systemic antibiotic other than macrolides were used in 10,885 (78.1%). The most commonly prescribed antibiotics were beta-lactams (72.2%), quinolones

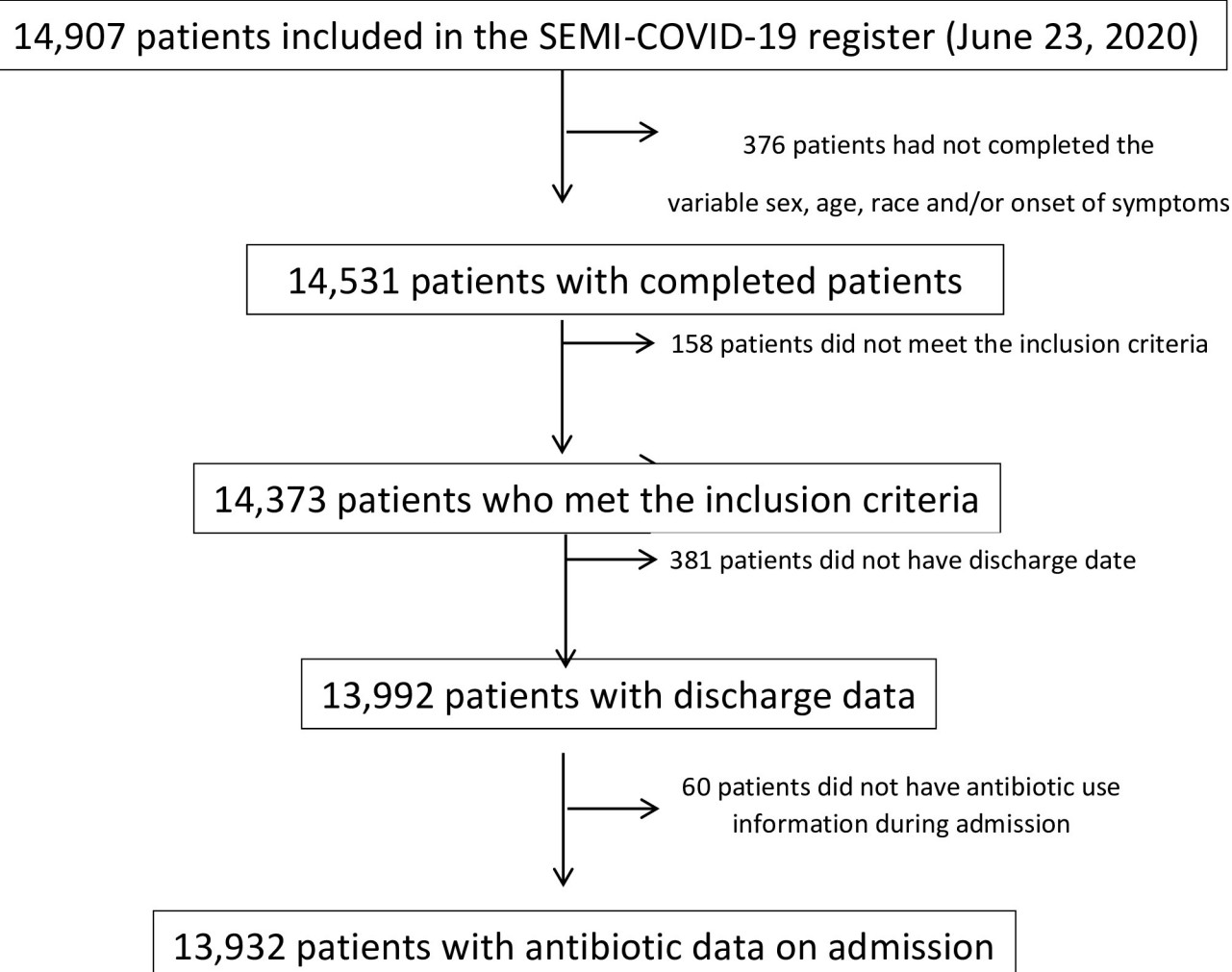

14,907 patients included in the SEMI-COVID-19 register (June 23, 2020)

376 patients had not completed the variable sex, age, race and/or onset of symptoms

14,531 patients with completed patients

158 patients did not meet the inclusion criteria

14,373 patients who met the inclusion criteria

381 patients did not have discharge date

13,992 patients with discharge data

60 patients did not have antibiotic use information during admission

13,932 patients with antibiotic data on admission

**Fig 1. Patient flowchart.**

(13.4%), linezolid (2.2%), glycopeptides (1.6%), co-trimoxazole (0.6%), and tetracyclines (0.6%). The rest of the antibiotics accounted for less than 0.3% of patients each.

### Criteria for antibiotic prescription

Of all patients, 52.4% met at least one criterion for the use of antibiotics. The most common criteria were unilateral alveolar infiltrate (17.5%), cough with purulent expectoration (15.5%), negative SARS-CoV-2 PCR (11.9%), respiratory bacterial co-infection and/or superinfection (10.9%), sepsis (6.2%), procalcitonin equal to or greater than 0.5 ng/mL (5.7%), significant pleural effusion (3.0%), shock (4.5%), and positive pneumococcal and/or legionella urine antigens (1.5%). The presence of superinfections other than respiratory superinfections was rare, including venous catheter-related bacteremia (0.8%), urinary tract infection (0.8%), abdominal infection (0.3%), skin and soft tissue infection (0.1%), other infections (0.1%). Table 1 summarizes the criteria for antibiotic prescription.

### Inappropriate antibiotic prescription

In total, non-macrolide antibiotics were not prescribed in 3,047 patients (21.6%), whereas they were appropriately prescribed in 6,116 patients (43.9%), and inappropriately prescribed in

**Table 1. ATB prescription criteria in COVID-19 patients.**

| ATB prescription criteria | Total (n = 13932) | Appropriate ATB (n = 6116) | No ATB (n = 3047) |
|---|---|---|---|
| Purulent expectoration | 15.5% (2157) | 29.8% (1816) | 11.2% (341) |
| Sepsis | 6.2% (853) | 13.0% (789) | 2.1% (64) |
| Shock | 4.5% (625) | 9.6% (587) | 1.3% (38) |
| Unilateral alveolar infiltrate | 17.5% (2411) | 33.0% (2007) | 13.7% (404) |
| Significant pleural effusion | 3.0% (413) | 5.7% (344) | 2.3% (69) |
| CT no compatible with COVID-19 | 0.6% (88) | 1.0% (63) | 0.8% (25) |
| Negative first SARS-CoV2 PCR | 12.1% (1660) | 21.9% (1320) | 11.3% (340) |
| PCT equal or greater than 0.5 ng/mL | 5.7% (797) | 11.6% (709) | 2.9% (88) |
| Respiratory superinfection | 10.9% (1508) | 23.6% (1439) | 2.3% (69) |
| Other superinfections | 2.2% (314) | 5.1% (311) | 3 (0.1%) |
| Any criteria | 52.4% (7294) | 100% (6116) | 38.7% (1178) |

ATB: antibiotic. CT: Computed Tomography. PCT: Procalcitonin.

4,769 patients (34.2%). Accordingly, 43.8% of antibiotic prescriptions were considered inappropriate. The epidemiological, clinical, analytical, and radiological characteristics of patients with appropriate vs inappropriate antibiotic prescribing and patients who were not prescribed antibiotics vs patients with inappropriate prescribing are summarized in Tables 2 and 3, respectively, as well as the risk factors, with adjusted OR for inappropriate prescribing.

Out of a total of 1078 critically ill patients who were admitted to ICU units, no antibiotics were prescribed in 29 patients (2.7%), whereas they were appropriately prescribed in 833 patients (77.3%), and inappropriately prescribed in 216 patients (20.0%).

## Antibiotic prescription over time

A total of 11,611 (83.3%) patients were admitted in February or March 2020 (group 1) whereas 2,321 patients (16.7%) were admitted later (group 2). In the first group, non-macrolides antibiotics were used in 9,231 patients (79.5%) compared to 1,654 (71.3%) in the second group admitted after March, a statistically significant difference ($p < .001$). However, an indication for antibiotics was less common in those admitted in the first group compared to the second group (51.8% vs 55.1%, $p = .003$). Thus, inappropriate antibiotic use was less common in the second group (28.0% vs 35.5%, $p < .001$). Fig 2 shows antibiotic prescription variation over time.

## Potential adverse effects for antibiotic prescription

The occurrence of complications potentially resulting from pharmacologic prescription was more frequent in patients with antibiotics (19.6% vs 10.5%, OR 2.07, 95% CI 1.82–2.35, p<0.001). Due to the design of the register, it was not possible to know if acute renal injury (AKI) was present at admission and, therefore, before antibiotic prescription. However, if we exclude AKI from other complications, patients with antibiotics prescription were more likely to have drug-related complication that patients without them (4.9% vs 2.7%, OR 1.84, 95% CI 1.45–2.32, p<0,001). The main complications potentially resulting from the use of antibiotics are summarized in Table 4. The presence of complications was similar in patients with appropriate and inappropriate prescriptions.

## Discussion

In this study, we aimed to analyze inappropriate antibiotic prescribing in patients with COVID-19 as well as its risk factors. Inappropriate antibiotic prescribing was very high in our

**Table 2. Epidemiological, clinical and analytical characteristics according to the appropriate or inappropriate prescription of ATB.**

| Variable | Univariant | | | Multivariant | | Missing (10885) |
|---|---|---|---|---|---|---|
| | Appropriate ATB (6116) | Inappropriate ATB (4769) | p | OR (95% CI) | p | |
| Epidemiological | | | | | | |
| February-March admission (vs later) | 83.6% (5113) | 86.3% (4118) | <0.001 | 1.27 (1.12–1.43) | <0.001 | 0 |
| Age | 68.7 (57.7–77.2) | 66.0 (56.5–77.2) | <0.001 | 0.99 (0.98–1.00) | <0.001 | 0 |
| Sex (male) | 60.3% (3689) | 56.2% (2684) | <0.001 | 0.87 (0.78–0.98) | 0.031 | 0 |
| Charlson Index | 1 (0–2) | 0 (0–1) | <0.001 | | | 288 |
| Age-Adjusted Charlson Index | 3 (2–5) | 2 (1–4) | <0.001 | 0.87 (0.77–0.97) | 0.018 | 288 |
| Alcoholism | 5.6% (333) | 3.8% (172) | <0.001 | | | 397 |
| Smoking | 27.7% (1610) | 24.3% (1095) | <0.001 | | | 561 |
| Severe dependence | 9.1% (550) | 5.2% (244) | <0.001 | 1.03 (0.64–1.66) | 0.879 | 148 |
| Arterial hypertension | 53.8% (3286) | 49.2% (2342) | <0.001 | 0.99 (0.83–1.17) | 0.899 | 17 |
| Obesity | 22.3% (1233) | 22.9% (942) | 0.643 | | | 1063 |
| SOT | 1.3% (80) | 1.2% (59) | 0.148 | | | 114 |
| IS | 4.0% (244) | 3.2% (151) | 0.468 | | | 37 |
| Coronary disease | 6.6% (403) | 5.5% (262) | 0.017 | | | 17 |
| Heart failure | 9.0% (547) | 5.4% (257) | <0.001 | 0.94 (0.69–1.26) | 0.684 | 23 |
| COPD | 8.6% (527) | 5.6% (267) | <0.001 | 1.04(0.77–1.39) | 0.799 | 22 |
| Asthma | 7.7% (470) | 6.3% (298) | 0.004 | | | 24 |
| Stroke | 3.2% (196) | 2.3% (109) | 0.004 | | | 22 |
| Cognitive impairment | 12.1% (736) | 8.4% (402) | <0.001 | 0.98 (0.72–1.33) | 0.901 | 20 |
| Chronic kidney failure | 7.7% (474) | 4.3% (204) | <0.001 | 0.77 (0.54–1.08) | 0.135 | 29 |
| Active cancer | 6.8% (415) | 5.5% (261) | 0.005 | | | 18 |
| Diabetes mellitus | 14.7% (895) | 14.2% (678) | 0.519 | | | 25 |
| AID | 2.4% (145) | 2.2% (107) | 0.660 | | | 31 |
| AIDS | 0.3% (20) | 0.2% (8) | 0.149 | | | 43 |
| Non-AIDS HIV | 0.8% (46) | 0.6% (27) | 0.235 | | | 43 |
| At least one comorbidity | 82.3% (5035) | 76.7% (3657) | <0.001 | 0.81 (0.68–0.97) | 0.022 | 0* |
| Clinical symptoms | | | | | | |
| Dry cough | 46.6% (2838) | 72.0% (3424) | <0.001 | 3.59 (3.13–4.13) | <0.001 | 41 |
| Arthromyalgia | 28.3% (1711) | 32.9% (1543) | <0.001 | 0.91 (0.78–1.06) | 0.238 | 154 |
| Ageusia | 6.3% (375) | 8.0% (370) | 0.001 | | | 352 |
| Anosmia | 5.5% (323) | 7.3% (338) | <0.001 | | | 355 |
| Asthenia | 43.1% (2589) | 45.5% (2130) | 0.014 | | | 192 |
| Odynophagia | 9.0% (539) | 10.1% (473) | 0.047 | | | 222 |
| Headache | 10.0% (598) | 12.1% (567) | 0.001 | | | 206 |
| Fever | 64.7% (3945) | 67.0% (3184) | 0.014 | 1.11 (1.01–1.23) | 0.031 | 40 |
| Dyspnea | 61.1% (3719) | 57.0% (2706) | <0.001 | 0.99 (0.70–1.40) | 0.984 | 49 |
| Diarrhea | 20.9% (1268) | 25.9% (1220) | <0.001 | 1.15 (0.79–1.67) | 0.457 | 101 |
| Abdominal pain | 6.0% (363) | 6.6% (308) | 0.255 | | | 150 |
| Crackles | 54.6% (3243) | 53.7% (2492) | 0.387 | | | 307 |
| Flu-like symptoms | 4800 (78.5%) | 4549 (95.4%) | <0.001 | 3.2 (1.67–6.13) | <0.001 | 0* |
| Laboratory and image test | | | | | | |
| pH | 7.45 (7.42–7.49) | 7.45 (7.42–7.48) | 0.316 | | | 2370 |
| pO2 (%) | 66 (55–76) | 65 (57–75) | 0.251 | | | 2886 |
| PaFi | 281 (235–332) | 294 (249–333) | <0.001 | 1,00 (1,00–1,04) | 0,110 | 2990 |
| Hemoglobin (g/dL) | 13.6 (12.3–14.7) | 14.1 (12.8–14.9) | <0.001 | 1.06 (0.97–1.16) | 0.140 | 118 |
| Plateles (x10$^9$/L) | 199 (154–256) | 195 (149–268) | 0.953 | | | 118 |

*(Continued)*

**Table 2.** (Continued)

| Variable | Univariant | | | Multivariant | | Missing (10885) |
|---|---|---|---|---|---|---|
| | Appropriate ATB (6116) | Inappropriate ATB (4769) | p | OR (95% CI) | p | |
| Leucocytes (x10⁹/L) | 6.4 (5.0–8.6) | 6.5 (5.1–8.6) | <0.001 | 1.00 (1.00–1.00) | 0.912 | 118 |
| Lymphocytes (x10⁹/L) | 0.8 (0.6–1.1) | 0.9 (0,6–1.2) | 0.191 | | | 118 |
| Neutrophils (x10⁹/L) | 3.2 (2.4–4.9) | 3.7 (2.6–4.8) | <0.001 | | | 118 |
| CRP (mg/L) | 71 (15–146) | 50 (12–115) | <0.001 | 0.99 (0.98–1.01) | 0.220 | 768 |
| LDH (U/L) | 346 (245–512) | 341 (261–444) | <0.001 | | | 1044 |
| Ferritin (microg/L) | 662 (324–1121) | 672 (359–1507) | 0.363 | | | 3019 |
| IL-6 (ng/L) | 32.8 (12.5–74.0) | 24.0 (6.5–66.0) | 0.008 | 1.00 (0.99–1.01) | 0.500 | 5335 |
| D-Dimer (ng/mL) | 0.77 (0.43–1.38) | 0.59 (0.36–1.06) | <0.001 | 1.00 (1.00–1.00) | 0.361 | 1128 |
| Interstitial infiltrate | 58.1% (3530) | 71.4% (3333) | <0.001 | 1.40 (1.30–1.54) | <0.001 | 78 |

Quantitative variables are expressed as median (interquartile range). Qualitative variables as percentage (total number). Variables that achieved statistically significant and with a clinically relevant difference between groups were included in a single-step multivariate logistic regression model. Adjusted OR for inappropriate prescription are provided for all variables included in the model. ATB: antibiotic. OD: Odds Ratio. CI: Confidence Interval. SOT: Solid Organ Transplant. IS: immunosuppression. COPD: Chronic Obstructive Pulmonary Disease. AID: Autoimmune Disease. AIDS: Acquired Human Immunodeficiency Syndrome. HIV: Human Immunodeficiency Virus. IL-6: Interleukin 6

patients. Younger age; less comorbidity; and the presence of dry cough, flu-like symptoms, fever, bilateral interstitial infiltrates, and increased C-Reactive Protein (CRP) levels were independently associated with inappropriate prescribing.

The use of antibiotics in our patients was common, and accounts for more than three quarters of patients. This percentage is similar to what has been found in other cohorts [21–23] and meta-analyses [3, 4, 6]. The percentage of antibiotic use was especially high in ICU patients, although the inappropriate use in this setting was relatively low (20.0%) The ample use of antibiotics contrasts with the low incidence of bacterial co-infection or superinfection found. Only 10% of patients had confirmed pulmonary superinfection while 2% had superinfection of another origin, the most common being venous catheter-related bacteremia and urinary tract infection. Again, these data are in line with those described previously by other authors [4, 22, 24], with higher percentages described in critical patients [25, 26], which could justify the higher use of antibiotic use that we found in those patients. It should be noted that, due to the design of our database, we were unable to distinguish between community-acquired pulmonary co-infection and nosocomial superinfection, though the latter [21, 22, 25, 27].

One half of our patients met one or more appropriate antibiotic use criteria. The criteria selected for its use in patients with SARS-CoV-2 infection in this work are similar to those proposed by other authors in several publications [10, 16, 28, 29]. With this in mind, antibiotics were used inappropriately in more than a third of all patients. Independent risk factors for inappropriate prescribing were younger age; less comorbidity; and the presence of dry cough, fever, flu-like symptoms, and bilateral interstitial infiltrates. Independent risk factors for inappropriate prescribing vs no antibiotic prescribing were younger age; presence of dry cough, fever, dyspnea, flu-like symptoms, or higher CRP levels. In both cases, the factors that were most strongly linked to inappropriate prescribing were dry cough or flu-like symptoms. We also detected a lower percentage of inappropriate antibiotic prescribing in patients who were admitted to the hospital after March 2020, which can perhaps be explained by healthcare professionals' greater knowledge of the disease.

The use of antibiotics in these patients is not without risk. In our series, we found more adverse drug reactions in patients receiving antibiotics. AKI, pharmacological

**Table 3. Epidemiological, clinical and analytical characteristics according to non-prescription or inappropriate prescribing of ATB.**

| Variable | Univariant | | | Multivariant | | Missing (n = 7816) |
|---|---|---|---|---|---|---|
| | No ATB (3047) | Inappropriate ATB (4769) | p | OR (95% CI) | p | |
| **Epidemiological** | | | | | | |
| February-March admission (vs later) | 78.1% (2380) | 86.3% (4118) | <0.001 | 1.54 (1.18–2.00) | 0.002 | 0 |
| Age | 66.1 (53.0–77,5) | 66.0 (56.5–77.2) | <0.001 | 0.98 (0.97–0.99) | 0.014 | 0 |
| Sex (male) | 52.0% (1583) | 56.2% (2677) | <0.001 | 0.91 (0.71–1.16) | 0.455 | 9 |
| Charlson Index | 1 (0–2) | 0 (0–1) | 0.339 | | | 197 |
| Age-Adjusted Charlson Index | 3 (1–5) | 2 (1–4) | 0.011 | 0.99 (0.91–1.06) | 0.807 | 197 |
| Alcoholism | 4.1% (123) | 3.8% (172) | 0.401 | | | 269 |
| Smoking | 20.7% (607) | 24.3% (1095) | <0.001 | | | 376 |
| Severe dependence | 6.5% (193) | 5.2% (244) | 0.053 | | | 132 |
| Arterial hypertension | 56.3% (3286) | 49.2% (2342) | 0.015 | 0.91 (0.68–1.21) | 0.536 | 11 |
| Obesity | 18.1% (1233) | 21.9% (942) | <0.001 | | | 686 |
| SOT | 1.1% (33) | 1.2% (59) | 0.454 | | | 112 |
| IS | 3.1% (94) | 3.2% (151) | 0.902 | | | 26 |
| Coronary disease | 5.3% (161) | 5.5% (262) | 0.694 | | | 10 |
| Heart failure | 6.1% (187) | 5.4% (257) | 0.162 | | | 14 |
| COPD | 5.4% (163) | 5.6% (267) | 0.644 | | | 11 |
| Asthma | 8.1% (245) | 6.3% (298) | 0.003 | | | 17 |
| Stroke | 2.8% (84) | 2.3% (109) | 0.192 | | | 10 |
| Cognitive impairment | 8.6% (261) | 8.4% (402) | 0.814 | | | 22 |
| Chronic kidney failure | 5.2% (157) | 4.3% (204) | 0.082 | | | 11 |
| Active cancer | 6.0% (184) | 5.5% (261) | 0.292 | | | 7 |
| Diabetes mellitus | 12.0% (366) | 14.2% (678) | 0.006 | | | 16 |
| AID | 2.3% (70) | 2.2% (107) | 0.875 | | | 22 |
| AIDS | 0.4% (12) | 0.2% (8) | 0.083 | | | 27 |
| Non-AIDS HIV | 0.8% (24) | 0.6% (27) | 0.232 | | | 27 |
| At least one comorbidity | 75.1% (2289) | 76.7% (3657) | 0.015 | 1.43 (1,05–1.94) | 0.025 | 0 |
| **Clinical symptoms** | | | | | | |
| Dry cough | 59.7% (1815) | 72.0% (3424) | <0.001 | 2.51 (1.94–3.26) | <0.001 | 25 |
| Arthromyalgia | 28.9% (874) | 32.9% (1543) | <0.001 | | | 102 |
| Ageusia | 8.9% (266) | 8.0% (370) | 0.188 | | | 211 |
| Anosmia | 8.0% (238) | 7.3% (338) | 0.316 | | | 216 |
| Asthenia | 40.6% (1226) | 45.5% (2130) | <0.001 | | | 110 |
| Odynophagia | 10.0% (301) | 10.1% (473) | 0.814 | | | 125 |
| Headache | 12.7% (384) | 12.1% (567) | 0.442 | | | 122 |
| Fever | 55.5% (1688) | 67.0% (3184) | <0.001 | 1.33 (1.13–1.56) | 0.001 | 25 |
| Dyspnea | 51.6% (1566) | 57.0% (2706) | <0.001 | 1.31 (1.04–1.69) | 0.044 | 34 |
| Diarrhea | 23.9% (722) | 25.9% (1220) | 0.048 | | | 64 |
| Abdominal pain | 6.9% (209) | 6.6% (308) | 0.522 | | | 97 |
| Crackles | 46.8% (1388) | 53.7% (2492) | <0.001 | 0.89 (0.70–1.13) | 0.358 | 216 |
| Flu-like symptoms | 90.2% (2747) | 95.4% (4549) | <0.001 | 2.70 (1.75–4.17) | <0.001 | 0 |
| **Laboratory and image test** | | | | | | |
| pH | 7.44 (7.41–7.47) | 7.45 (7.42–7.48) | <0.001 | | | 1952 |
| pO2 (%) | 68 (59–81) | 65 (57–75) | <0.001 | 0.99 (0.98–1.01) | 0.544 | 2370 |
| PaFi | 304 (253–361) | 294 (249–333) | <0.001 | | | 2450 |
| Hemoglobin (g/dL) | 13,9 (12.7–15.0) | 14,1 (12.8–14.9) | 0.009 | | | 156 |
| Platelets (x$10^9$/L) | 194 (153–249) | 195 (149–268) | 0.050 | | | 156 |

*(Continued)*

**Table 3.** (Continued)

| Variable | Univariant | | | Multivariant | | Missing (n = 7816) |
|---|---|---|---|---|---|---|
| | No ATB (3047) | Inappropriate ATB (4769) | p | OR (95% CI) | p | |
| Leucocytes (x10⁹/L) | 6.1 (4.7–8.1) | 6.5 (5.1–8.6) | 0.183 | | | 156 |
| Lymphocytes (x10⁹/L) | 1.0 (0.7–1.4) | 0.9 (0.6–1.2) | <0.001 | 1.00 (1.00–1.00) | 0.750 | 156 |
| Neutrophils (x10⁹/L) | 4.3 (3.0–6.2) | 3.7 (2.6–4.8) | 0.007 | | | 156 |
| CRP (mg/L) | 32.7 (8.7–88.0) | 50 (12–115) | <0.001 | 1.01 (1,00–1.01) | 0.001 | 1776 |
| LDH (U/L) | 296 (230–401) | 341 (261–444) | <0.001 | 1.00 (0.99–1.00) | 0.560 | 1776 |
| Ferritin (microg/L) | 487 (222–1090) | 672 (359–1507) | <0.001 | 1.00 (1.00–1.00) | 0.581 | 1776 |
| IL-6 (ng/L) | 27.5 (10.7–54.3) | 24.0 (6.5–66.0) | 0.272 | | | 6810 |
| D-Dimer (ng/mL) | 0.61 (0.36–1.12) | 0.59 (0.36–1.06) | 0.114 | | | 1709 |
| Interstitial infiltrate | 58.9% (1741) | 71.4% (3333) | <0.001 | 1.02 (0.88–1.23) | 0.821 | 127 |

Quantitative variables are expressed as median (interquartile range). Qualitative variables as percentage (total number). Variables that achieved statistically significant and with a clinically relevant difference between groups were included in a single-step multivariate logistic regression model. Adjusted OR for inappropriate prescription are provided for all variables included in the model. ATB: antibiotic. OD: Odds Ratio. CI: Confidence Interval. SOT: Solid Organ Transplant. IS: immunosuppression. COPD: Chronic Obstructive Pulmonary Disease. AID: Autoimmune Disease. AIDS: Acquired Human Immunodeficiency Syndrome. HIV: Human Immunodeficiency Virus. IL-6: Interleukin 6

hypertransaminasemia, drug-induced diarrhea, and candidiasis were more common in patients who received them. In addition, we identified an incidence of 2.5 allergic reactions per 1000 prescriptions, a similar figure to what has been reported in the recent literature [30]. We also identified an incidence of 2.7 cases CD infection per 1000 prescriptions, a higher incidence than what has been found in COVID-19 patients by other authors [31]. One of the most feared secondary effect of inappropriate antibiotic use is an increase in microbial resistance [2,

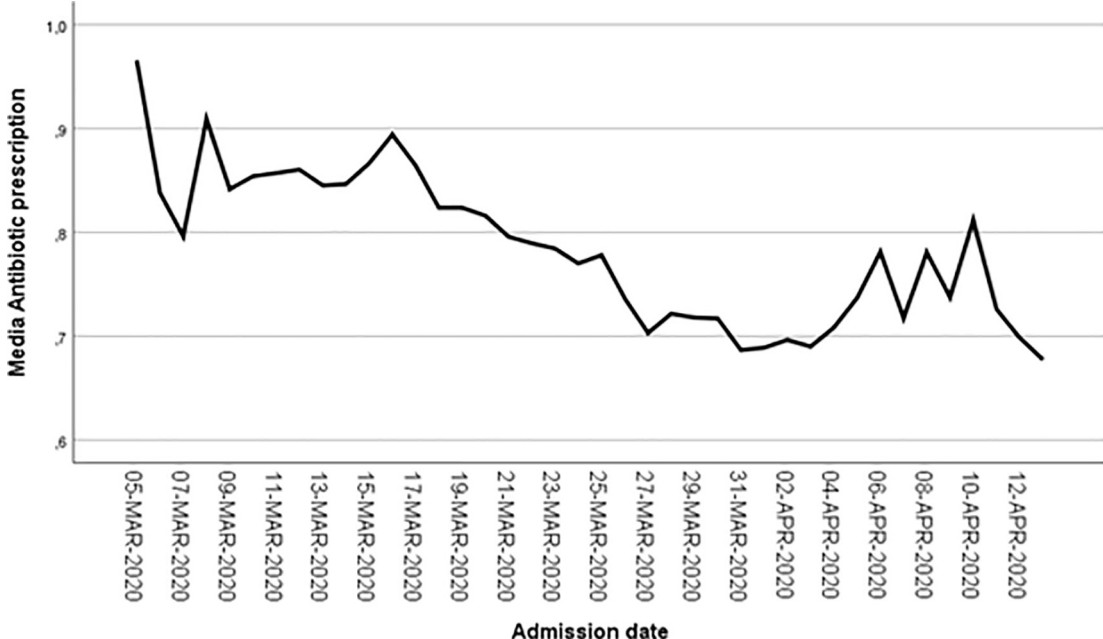

**Fig 2. Antibiotic prescription variation over time.** Abscissa axis corresponds to admission date. Ordinate axis corresponds to the percentage of patients with antibiotic prescription (scheme 0.9 equals to 90%). Data are shown for the period with more than 20 admissions per day.

**Table 4. Comparison of incidence of complications potentially associated with ATB.**

| Complication | Without ATB (n = 3047) | With ATB (n = 10885) | P | Appropriate ATB (n = 4769) | Inappropriate ATB (n = 6116) | p |
|---|---|---|---|---|---|---|
| **Hypertransaminasemia** | 1.0% (29) | 2.1% (226) | <0.001 | 2.1% (129) | 2.0% (97) | 0.787 |
| **Iatrogenic diarrhea** | 0.8% (25) | 1.3% (137) | 0.025 | 1.0% (62) | 1.6% (75) | 0.012 |
| **AKI** | 8.1% (246) | 15.6% (1690) | <0.001 | 19.9% (1211) | 10.1% (479) | <0.001 |
| **Allergic reaction to ATB** | 0.1% (3) * | 0.2% (25) | 0.072 | 0.2% (12) | 0.3% (13) | 0.426 |
| **Prolongated QT** | 0.5% (14) | 0.3% (37) | 0.334 | 0.4% (22) | 0.4% (15) | 0.688 |
| **Neutropenia** | 0.2% (5) | 0.2% (18) | 0.988 | 0.2% (13) | 0.1% (5) | 0.235 |
| **Thrombocytopenia** | 0.2% (5) | 0.3% (38) | 0.068 | 0.4% (26) | 0.3% (12) | 0.142 |
| *Clostridioides difficile* | <0.1% (2) * | 0.3% (28) | 0.026 | 0.3% (18) | 0.2% (10) | 0.449 |
| **Candidemia** | <0.1% (2) | 0.1% (14) | 0.181 | 0.2% (14) | 0 | 0.002 |
| **Candidiasis** | <0.1% (2) | 0.4% (39) | 0.003 | 0.4% (23) | 0.3% (16) | 0.750 |
| **Any AR (excluding AKI)** | 2.7% (83) | 4.9% (533) | <0.001 | 4.9% (300) | 4.9% (233) | 0.964 |
| **Any AR (including AKI)** | 10.5% (321) | 19.6% (2134) | <0.001 | 23.7% (1448) | 14.4% (686) | <0.001 |

ATB: Antibiotics. AKI: Acute Kidney Injury, AR: adverse reaction.

*Effects that were attached to macrolide.

7, 32]. It is well known that the spread of multidrug-resistant bacteria is closely related to antibiotic exposure [2, 32, 33]. Although due to limitations in the database we could not analyze antibiotic resistance, we can speculate that the overuse of antibiotic and inappropriately prescribed antibiotics in COVID-19 patients can induce an increase in antibiotic resistance, which have already been noted by some authors [34].

Therefore, by inappropriately prescribing antibiotics, we are exposing patients with SARS-CoV-2 infection to pharmacological toxicity and increased risk of morbidity despite the fact that no benefits have been proven, even in critical patients [35]. Inappropriate antibiotic prescribing may be due to multiple factors, such as an overload of the healthcare system, confusion with bacterial pneumonia, etc. Moreover, several local management protocols in March and April 2020 advised physicians to prescribed empirical antibiotics (such as cephalosporins) to nearly all patients regardless of whether there was an indication. Those protocols must be changed and the recommendation to prescribe empirical antibiotics in absence of a possible of bacterial superinfection must be removed.

The need to implement specific criteria for antibiotic use in COVID-19 patients has previously been emphasized [36] and indeed, clinical guidelines for its prescription in COVID-19 patients [37]. In this document, the main recommendation is to restrict the use of these drugs, especially at the time of admission, when bacterial infections are less common [22]. It even recommends early suspension of the antibiotic courses that may have been started in the emergency department. It may be challenging to discern which patients warrant antibiotic prescription and in which patients antibiotic use may be inappropriate. It is possible that our work could help identify patients who have been inappropriately prescribed. Accordingly, young patients and those without comorbidity who are prescribed antibiotics for dry cough, fever, flu-like symptoms, interstitial infiltrates, or increased CRP may be receiving them inappropriately and, in the absence of other data indicating their use, suspension of treatment could be considered. Another possible action should be to improve formation regarding appropriate antibiotic therapy and antimicrobial therapy and stewardship principles, both in general setting and specifically in COVID-19 patients, as has been shown previously by other authors that those aspects are poorly addressed in medical training programs [38].

Our work is based on a large, multicenter cohort and has the strengths inherent to these types of works: appropriate representation of different regions, which reduces biases of local

origin and increases external validity, as well as a large sample size, which provides statistical power. However, the study also has limitations. First, the main limitation is that although the database contains data on more than 300 variables, it was not specifically designed to analyze inappropriate antibiotic prescribing. Therefore, some important variables for determining whether or not the prescription was appropriate (such as antimicrobial spectrum, start time, duration of the course, etc.) were not available. We were not able to separately analyze antibiotic courses that began at the time of hospital admission versus those initiated later. Second, the data were collected by a large number of researchers from different centers, which could have led to heterogeneity in data collection, especially in the "other complications" variable and in the identification of bacterial complications. Third, our study is observational in nature, which prevents us from determining causal relationships. Fourth, our criteria of appropriate antibiotic prescribing are based in scarce low-quality evidence and it may be inadequate. However, our criteria were carefully selected based on the most recent literature evidence available. Although our selected criteria for appropriate antibiotic use may be too inclusive, until there is published more evidence on antibiotic use in COVID-19 patients (which is urgently needed), they should be considered as valid. Finally, due to limits in the allowed analysis of the database, we could not assess the association between antibiotic prescription and outcome, including mortality or readmissions. However, previous data shows that there is little or no benefic in their use without evidence of bacterial infection [21].

In conclusion, inappropriate use of antibiotics in our patients was a common phenomenon. Lower age, less comorbidity, the presence of dry cough, flu-like symptoms, fever, bilateral interstitial infiltrates and increased CRP were independently associated with inappropriate prescription. Less inappropriate prescription were detected in patients admitted after March. Widespread antibiotic prescribing carries an increased risk of adverse reaction and probably other unwanted effects (such as possible increased bacterial resistances), without benefit. It is therefore essential to integrate antibiotic use optimization programs in patients with SARS-CoV2 infection. More research is needed to identify patients which warrant antibiotic prescription.

## Supporting information

**S1 File. List of the SEMI-COVID-19 network members.**
(DOCX)

**S1 Annex. Examples of local protocols that included recommendation for macrolide use due to its supposed antiviral and immunomodulatory effect.**
(DOCX)

## Acknowledgments

We thank all researchers involved in the SEMI-COVID-19 Network Group for their efforts. A list of all members of the group is available in S1 File. We also thank the SEMI-COVID-19, S&H Medical Science Service Registry Coordination Center for its data quality control and logistical and administrative support.

## Author Contributions

**Conceptualization:** Jorge Calderón-Parra, Antonio Muiño-Miguez, Alejandro D. Bendala-Estrada, Antonio Ramos-Martínez, Elena Muñez-Rubio.

**Data curation:** Jorge Calderón-Parra.

**Formal analysis:** Jorge Calderón-Parra.

**Investigation:** Jorge Calderón-Parra, Antonio Ramos-Martínez, Elena Muñez-Rubio, José Manuel Casas Rojo, Jesús Millán Núñez-Cortés.

**Methodology:** Jorge Calderón-Parra, Antonio Ramos-Martínez.

**Project administration:** José Manuel Casas Rojo, Jesús Millán Núñez-Cortés.

**Supervision:** Antonio Ramos-Martínez.

**Validation:** Jorge Calderón-Parra.

**Writing – original draft:** Jorge Calderón-Parra, Antonio Ramos-Martínez.

**Writing – review & editing:** Jorge Calderón-Parra, Antonio Muiño-Miguez, Alejandro D. Bendala-Estrada, Antonio Ramos-Martínez, Elena Muñez-Rubio, Eduardo Fernández Carracedo, Javier Tejada Montes, Manuel Rubio-Rivas, Francisco Arnalich-Fernandez, Jose Luis Beato Pérez, Jose Miguel García Bruñén, Esther del Corral Beamonte, Paula Maria Pesqueira Fontan, Maria del Mar Carmona, Rosa Fernández-Madera Martínez, Andrés González García, Cristina Salazar Mosteiro, Carlota Tuñón de Almeida, Julio González Moraleja, Francesco Deodati, María Dolores Martín Escalante, María Luisa Asensio Tomás, Ricardo Gómez Huelgas, José Manuel Casas Rojo, Jesús Millán Núñez-Cortés.

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
