## [Decision Letter · Decision Letter 0]

14 Apr 2021

PONE-D-21-09419

Inappropriate antibiotic use in the COVID-19 era: factors associated with inappropriate prescribing and secondary complications. Analysis of the registry SEMI-COVID.

PLOS ONE

Dear Dr. Parra,

Thank you for submitting your manuscript to PLOS ONE. After careful consideration, we feel that it has merit but does not fully meet PLOS ONE’s publication criteria as it currently stands. Therefore, we invite you to submit a revised version of the manuscript that addresses the points raised during the review process.

We look forward to receiving your revised manuscript.

Kind regards,

Francesco Di Gennaro

Academic Editor

PLOS ONE

Journal Requirements:

2. Thank you for including your ethics statement: The registry has the approval of the Ethics and Research Committee of the Province of Malaga. Patients were asked for informed consent. When it was not possible to obtain it for biosecurity reasons or because the patient was already discharged, it was collected verbally, leaving evidence in their medical history.

Please provide additional details regarding participant consent. In the ethics statement in the Methods and online submission information, please ensure that you have specified what type you obtained (for instance, written or verbal, and if verbal, how it was documented and witnessed). If your study included minors, state whether you obtained consent from parents or guardians. If the need for consent was waived by the ethics committee, please include this information.

3, One of the noted authors is a group or consortium SEMI-COVID-19 Network in addition to naming the author group, please list the individual authors and affiliations within this group in the acknowledgments section of your manuscript. Please also indicate clearly a lead author for this group along with a contact email address.

4, Thank you for stating the following in the Funding Section of your manuscript:

This research received no external funding and “The APC was funded by the Concejelía de

Sanidad, Comunidad de Madrid.

The authors recieved no specific funding for this work.

Additional Editor Comments:

dear authors follow reviewer suggestions to improve your paper

Reviewers' comments:

Reviewer's Responses to Questions

**Comments to the Author**

1. Is the manuscript technically sound, and do the data support the conclusions?

Reviewer #1: Yes

Reviewer #2: Yes

2. Has the statistical analysis been performed appropriately and rigorously? 

Reviewer #1: Yes

Reviewer #2: Yes

3. Have the authors made all data underlying the findings in their manuscript fully available?

Reviewer #1: Yes

Reviewer #2: Yes

4. Is the manuscript presented in an intelligible fashion and written in standard English?

Reviewer #1: Yes

Reviewer #2: Yes

5. Review Comments to the Author

Reviewer #1: The study is well designed and included large patient pool for many centers. Indeed, there was and still is inappropriate antibiotic use in COVID-19 infection, specially it was rampant in the beginning of the pandemic. The methodology is explained in detail; patient selection criteria is made as much elaborated as possible. The result and Tables/Figures are used appropriately.

Few questions and suggestions:

1. The aim or the hypothesis of the study is not clear. It should be clearly outlined in abstract and at the end of background.

2. The primary outcome and secondary outcome should be mentioned. Without Primary and secondary outcomes, it is difficult to interpret authors findings. For example, MV analysis reported in abstract’s result section: February-March 2020 admission (OR 1.54, 95%CI 1.18- 2.00), age (OR 0.98, 95%CI 0.97-0.99), …elevated C-reactive protein levels (OR 1.01 for each mg/L increase, 95% CI 1.00- 1.01) is for inappropriate ATB in comparison to No ATB. (But from background it seems the main purpose or aim of this study is to see Appropriate vs Inappropriate ATB). If the aim and the primary and secondary outcomes are No ATB vs Inappropriate Antibiotics, it should be mentioned. Otherwise it is confusing to interpret the data.

3. “Inclusion criteria: e) all data on antibiotic use during hospitalization available.” The authors in Figure 1 reported that 60 patients were excluded as they did not have antibiotic data on admission. ALL DATA means from since admission to discharge?

This is in contrast to criteria of appropriate prescribing (line 163): … including respiratory bacterial coinfection (at admission time) or superinfection (later on admission) with microbial isolation. If those 60 patients were excluded based on no antibiotic data on admission, then how authors included other patients who got superinfection (later on admission) with microbial isolation (because they might have been not prescribed antibiotic on admission hence lacking antibiotic data on admission).

This needs to be clarified.

4. Mention Table 1 in the parenthesis around line 165-166 where appropriate.

5. In Table 2 and 3, “Inadequate ATB and Adequate ATB” should be replaced with “Appropriate Antibiotics and Inappropriate Antibiotics”. The units for po2, lymphocytes, neutrophils etc. should be mentioned.

6. The odds ratio in line 264 and 268: What were they controlled for? Table for regression analysis?

7. Discussion: The first paragraph should be adjusted after the #2 is addressed.

8. Conclusion: Better to make it shorter. The authors attempted to include too many information that are diluting the main take-home message. (Conclusion should be to the point discussing hypothesis and the result in shortest possible words).

Overall a very good research study. I recommend to consider for publication after revisions.

Reviewer #2: Herein Authors discussed the important concern of antibiotic overprescribing in course of COVID-19 pandemic. I believe that this is a very important topic, often underestimated.

Overall, the manuscript is worth for publication, after few minor revisions:

1) A little bit more information should be given regarding the current literature on secondary bacterial infections in COVID-19;

2) A few lines should be spent regarding the high incidence of inappropriate antibiotic prescription in patients transferred to ICU;

3) In my opinion more emphasis should be place regarding the important spreading of multidrug-resistant bacteria due to inappropriate antimicrobial use and lack of infection control procedures;

4) A few lines regarding the importance of knowledge and practice regarding antimicrobial therapy should be included. I suggest citing this work regarding this aspect (Di Gennaro F, et al. Italian young doctors' knowledge, attitudes and practices on antibiotic use and resistance: A national cross-sectional survey. J Glob Antimicrob Resist. 2020 Dec;23:167-173. doi: 10.1016/j.jgar.2020.08.022).

6. PLOS authors have the option to publish the peer review history of their article (what does this mean?). If published, this will include your full peer review and any attached files.

Reviewer #1: No

Reviewer #2: No

---

## [Author Response · Author response to Decision Letter 0]

21 Apr 2021

Reviewer #1: 

The study is well designed and included large patient pool for many centers. Indeed, there was and still is inappropriate antibiotic use in COVID-19 infection, specially it was rampant in the beginning of the pandemic. The methodology is explained in detail; patient selection criteria is made as much elaborated as possible. The result and Tables/Figures are used appropriately.

Few questions and suggestions:

1. The aim or the hypothesis of the study is not clear. It should be clearly outlined in abstract and at the end of background.

Thank you for the suggestion. We have clarify the objectives of the study both in abstract and in background section. We aimed to analyze antibiotic prescription in order to determine the proportion of patients with inappropriate prescription and describe its complications. 

2. The primary outcome and secondary outcome should be mentioned. Without Primary and secondary outcomes, it is difficult to interpret authors findings. For example, MV analysis reported in abstract’s result section: February-March 2020 admission (OR 1.54, 95%CI 1.18- 2.00), age (OR 0.98, 95%CI 0.97-0.99), …elevated C-reactive protein levels (OR 1.01 for each mg/L increase, 95% CI 1.00- 1.01) is for inappropriate ATB in comparison to No ATB. (But from background it seems the main purpose or aim of this study is to see Appropriate vs Inappropriate ATB). If the aim and the primary and secondary outcomes are No ATB vs Inappropriate Antibiotics, it should be mentioned. Otherwise it is confusing to interpret the data.

Thank you for the observation. We have now mentioned primary and secondary outcomes in the manuscript. Primary outcome was proportion of inappropriate antibiotic prescription and its risk factors compared to appropriate antibiotic. Secondary outcomes included risk factors vs no antibiotic, complications and inappropriate prescription proportion over time. 

3. “Inclusion criteria: e) all data on antibiotic use during hospitalization available.” The authors in Figure 1 reported that 60 patients were excluded as they did not have antibiotic data on admission. ALL DATA means from since admission to discharge?

This is in contrast to criteria of appropriate prescribing (line 163): … including respiratory bacterial coinfection (at admission time) or superinfection (later on admission) with microbial isolation. If those 60 patients were excluded based on no antibiotic data on admission, then how authors included other patients who got superinfection (later on admission) with microbial isolation (because they might have been not prescribed antibiotic on admission hence lacking antibiotic data on admission).

This needs to be clarified.

We have explain poorly this inclusion criteria. We meant that we include patients in which the use (or not use) of antibiotic was available. Explained in other way, we have no included those patients in which the variables related to antibiotic use were missing (either not fulfill in the database by the investigator of fulfilled as unknown). We have clarify this point.

4. Mention Table 1 in the parenthesis around line 165-166 where appropriate.

Thank you for the suggestion. However, we feel that table 1 suits better in the result section (it express the percentage of patients with each indication), while the lines 165-166 are in the methods section.

5. In Table 2 and 3, “Inadequate ATB and Adequate ATB” should be replaced with “Appropriate Antibiotics and Inappropriate Antibiotics”. The units for po2, lymphocytes, neutrophils etc. should be mentioned.

Thank you for notice this mistake. We have corrected the terms and mention the units. 

6. The odds ratio in line 264 and 268: What were they controlled for? Table for regression analysis?

These odds ratio are not controlled, they are performed with univariate regression analysis. In order to clarify the results, we have now eliminated these OR and left only the p-value, obtained by means of chi-square test.

7. Discussion: The first paragraph should be adjusted after the #2 is addressed.

We have now adjusted the first paragraph to mention the aim and primary outcome of the study.

8. Conclusion: Better to make it shorter. The authors attempted to include too many information that are diluting the main take-home message. (Conclusion should be to the point discussing hypothesis and the result in shortest possible words).

Thank you for the suggestion. We have now shortened the conclusion in order to include the main information about our primary and secondary outcomes. 

Overall a very good research study. I recommend to consider for publication after revisions.

Reviewer #2: 

Herein Authors discussed the important concern of antibiotic overprescribing in course of COVID-19 pandemic. I believe that this is a very important topic, often underestimated.

Overall, the manuscript is worth for publication, after few minor revisions:

1) A little bit more information should be given regarding the current literature on secondary bacterial infections in COVID-19;

 We have given a more information ofg current literature about bacterial superinfection in COVID-19 patients. 

2) A few lines should be spent regarding the high incidence of inappropriate antibiotic prescription in patients transferred to ICU;

Thank you for the suggestion. We now discuss the high prevalence of antibiotic use in ICU patients (second paragraph).

3) In my opinion more emphasis should be place regarding the important spreading of multidrug-resistant bacteria due to inappropriate antimicrobial use and lack of infection control procedures;

Thank you for the suggestion. We now discuss the risk of spreading of multidrug-resistant bacteria due to antibiotic overuse in these patients.

4) A few lines regarding the importance of knowledge and practice regarding antimicrobial therapy should be included. I suggest citing this work regarding this aspect (Di Gennaro F, et al. Italian young doctors' knowledge, attitudes and practices on antibiotic use and resistance: A national cross-sectional survey. J Glob Antimicrob Resist. 2020 Dec;23:167-173. doi: 10.1016/j.jgar.2020.08.022).

Thank you for the suggestion. We have now added a few lines regarding the importance of knowledge and formation in antimicrobial theraphy.

---

## [Decision Letter · Decision Letter 1]

26 Apr 2021

Inappropriate antibiotic use in the COVID-19 era: factors associated with inappropriate prescribing and secondary complications. Analysis of the registry SEMI-COVID.

PONE-D-21-09419R1

Dear Dr. Parra,

We’re pleased to inform you that your manuscript has been judged scientifically suitable for publication and will be formally accepted for publication once it meets all outstanding technical requirements.

Kind regards,

Francesco Di Gennaro

Academic Editor

PLOS ONE

Additional Editor Comments (optional):

congratulations

Reviewers' comments:

Reviewer's Responses to Questions

**Comments to the Author**

1. If the authors have adequately addressed your comments raised in a previous round of review and you feel that this manuscript is now acceptable for publication, you may indicate that here to bypass the “Comments to the Author” section, enter your conflict of interest statement in the “Confidential to Editor” section, and submit your "Accept" recommendation.

Reviewer #1: All comments have been addressed

Reviewer #2: All comments have been addressed

2. Is the manuscript technically sound, and do the data support the conclusions?

Reviewer #1: Yes

Reviewer #2: Yes

3. Has the statistical analysis been performed appropriately and rigorously? 

Reviewer #1: Yes

Reviewer #2: Yes

4. Have the authors made all data underlying the findings in their manuscript fully available?

Reviewer #1: Yes

Reviewer #2: Yes

5. Is the manuscript presented in an intelligible fashion and written in standard English?

Reviewer #1: Yes

Reviewer #2: Yes

6. Review Comments to the Author

Reviewer #1: Thank you for adding the primary and secondary outcomes. And also for making changes as per recommendations.

Reviewer #2: (No Response)

7. PLOS authors have the option to publish the peer review history of their article (what does this mean?). If published, this will include your full peer review and any attached files.

Reviewer #1: No

Reviewer #2: No

---

## [Editor Report · Acceptance letter]

28 Apr 2021

PONE-D-21-09419R1 

Inappropriate antibiotic use in the COVID-19 era: factors associated with inappropriate prescribing and secondary complications. Analysis of the registry SEMI-COVID. 

Dear Dr. Calderón-Parra:

I'm pleased to inform you that your manuscript has been deemed suitable for publication in PLOS ONE. Congratulations! Your manuscript is now with our production department. 

Kind regards, 

on behalf of

Dr. Francesco Di Gennaro 

Academic Editor

PLOS ONE